# Novel Variants in Sperm Mitochondrial Cytochrome C Oxidase II (*MT-CO2*) Gene Associated with Asthenozoospermia in Jordan

**DOI:** 10.3390/cimb47110901

**Published:** 2025-10-29

**Authors:** Mazhar Salim Al Zoubi, Razan N. AlQuraan, Asmaa Al-Smadi, Mohammad A. AlSmadi, Manal AbuAlArja, Almuthanna K. Alkaraki, Bahaa Al-Trad, Raed M. Al-Zoubi, Khalid Al-Batayneh

**Affiliations:** 1Department of Basic Medical Sciences, Faculty of Medicine, Yarmouk University, Irbid 21163, Jordan; asmaasmadi@gmail.com (A.A.-S.); manalessam12@yahoo.com (M.A.); 2Department of Biological Sciences, Faculty of Science, Yarmouk University, Irbid 21163, Jordan; razan.quraan2024@gmail.com (R.N.A.); alkaraki@yu.edu.jo (A.K.A.); bahaa.tr@yu.edu.jo (B.A.-T.); albatynehk@yu.edu.jo (K.A.-B.); 3Reproductive Endocrinology and IVF Unit, King Hussein Medical Center, Amman 11733, Jordan; m_alsmadi@yahoo.com; 4Surgical Research Section, Department of Surgery, Hamad Medical Corporation, Doha 2713, Qatar; ralzoubi@hamad.qa; 5Department of Chemistry, Jordan University of Science and Technology, Irbid 22110, Jordan; 6Department of Biomedical Sciences, College of Health Sciences, Qatar University, Doha 2713, Qatar

**Keywords:** *MT-CO2*, male infertility, asthenozoospermia, mtDNA

## Abstract

Background: Asthenozoospermia is defined as a condition in which the total motility of sperm in a semen sample is less than 40%. Due to impairing sperm motility, asthenozoospermia was linked to different mitochondrial DNA (mtDNA) alterations. The current study aimed to investigate the relationship between *MT-CO2* gene variants and the development of asthenozoospermia and male infertility in the Jordanian population. Materials and Methods: Semen samples were collected from 196 men, including 119 asthenozoospermic (infertile) and 77 normozoospermia (control), from the Royal Jordanian Medical Services in vitro fertilization (IVF) unit. The isolated mitochondrial DNA (mtDNA) was subjected to a polymerase chain reaction to amplify the *MT-CO2* gene. Genetic variants were screened using direct Sanger sequencing. Genotypes and allele frequencies between the case and control groups were compared by the chi-square test and Fisher’s exact test. Results: Three novel variants in the *MT-CO2* gene were identified in nine asthenozoospermic cases, including two missense variants (m.8222T>A and m.7997G>A) and one synonymous variant (m.7846 A>G). In addition, the current study reported twenty-three known substitutions. In particular, the rs1556423316 T>C variant showed a significant association with asthenozoospermic infertile men in the studied population (*p* < 0.05). Conclusion: The detected missense variants in the *MT-CO2* gene in asthenozoospermic infertile men underscore the important role of these variants in the development of asthenozoospermic male infertility.

## 1. Introduction

Infertility is defined as a condition characterized by the inability to successfully conceive a pregnancy following one year or more of sexual intercourse without the use of contraception [1]. According to the World Health Organization (WHO), infertility is a prevalent condition that affects many individuals in their reproductive years globally, affecting 48 to 186 million individuals, with both male and female factors as contributing causes [2,3,4,5]. This condition has significant implications for their societal and economic well-being.

Male infertility affects around 7% of men all over the world and has been linked to various variables, including anatomical, immunological, physical, or obstructive diseases, hormonal and environmental factors, and other undiscovered causes classified as idiopathic factors [6]. Genetic factors are thought to be responsible for over half of all infertility cases. For instance, chromosomal abnormalities, single-gene, and polygenic disorders were linked to male infertility [7,8].

For successful fertilization, sperm motility plays a vital role that enables the spermatozoa to migrate from the vaginal canal towards the fallopian tube and subsequently facilitates the oocyte penetration process.

Sperm mitochondria produce adenosine triphosphate (ATP) and supply the sperm with the required energy needed for the fast-forward progressive movement of sperm [9]. The human mitochondrial deoxyribonucleic acid (mtDNA) structure is unique and distinguishable. It does not possess histones or other DNA-binding proteins, which enables it to replicate effectively without relying on DNA repair mechanisms [9,10,11]. Simultaneously, oxidative phosphorylation generates reactive oxygen species (ROS), which can potentially harm mitochondrial and cellular proteins, lipids, and mtDNA, thereby disrupting the process of energy production [11].

Asthenozoospermia, a prevalent cause of male infertility, refers to the low or diminished motility of sperm, wherein sperm motility falls below the threshold of 40% [12,13,14]. Numerous mutations within mtDNA have been associated with the condition known as asthenozoospermia [15,16,17,18,19]. MtDNA encodes thirteen proteins that constitute integral components of the respiratory chain within the mitochondria. These proteins exhibit subcellular localization within prominent complexes as outlined below: Complex I is composed of seven subunits of Nicotinamide Adenine Dinucleotide Hydride (NADH) dehydrogenase, namely MT-ND1, MT- ND2, MT-ND3, MT-ND4, MT-ND4L, MT-ND5, and MT-ND6. Complex III consists of cytochrome B, while complex IV comprises three subunits: MT-CO1, MT-CO2, and MT-CO3, which are subunits of cytochrome oxidase. Lastly, Complex V contains MT-ATPase 6 and MT-ATPase 8 [20,21,22,23]. The mitochondrial genes *MT-ATPase 6*, *MT- ATPase 8*, *MT-CO2*, *MT-CO3*, *MT-CytB*, *MT-ND3*, *MT-ND4*, *MT-ND5*, and *MT-ND6* are significantly involved in the development of mature sperm, and the generation of forward movement through the flagellum following ejaculation [10,24].

Cytochrome c oxidase 2 (COX2) acts as the ultimate complex in eukaryotic oxidative phosphorylation within mitochondria. COX2 enzyme comprises three subunits in mammals and yeast that originate from mitochondrial DNA, along with an additional 8 to 11 subunits derived from nuclear genes [25,26].

The *MT-CO2* gene encodes the COX subunit II enzyme. This enzyme is a crucial transmembrane subunit of COX2 in humans. A few mutations in the *MT-CO2* gene have been documented in infertile patients with asthenozoospermia. These mutations can potentially impact the electron transfer process from reduced cytochrome c to molecular oxygen within the mitochondria [23,25]. In addition, several studies have demonstrated that mutations in *MT-CO2* lead to diminished oxidative phosphorylation efficiency, decreased ATP synthesis, and reduced sperm motility due to impaired flagellar movement [10,22].

The objective of the present study is to assess the relation between *MT-CO2* genetic variants in asthenozoospermia development and male infertility in the Jordanian population.

## 2. Materials and Methods

### 2.1. Semen Samples Collection and Analysis

Following the approval of the Institutional Review Board at Yarmouk University (Approval code: 2022/28, Approval date: 19 May 2022), a total of 119 individuals diagnosed with asthenozoospermia (characterized by sperm motility ranging from 0% to <40%) were included in the study. The participants’ ages ranged from 18 to 40 years. Semen samples were obtained from a cohort of 77 normozoospermic males, with a normal percentage of sperm motility exceeding 40%. These individuals were selected as a control group for comparison with infertile subjects. The Jordanian Royal Medical Services IVF unit at Prince Rashid Ben Al-Hasan Military Hospital collected seminal specimens.

Excluded individuals are those who engaged in alcohol consumption or cigarette smoking, and those who had varicocele and were above 40. Semen samples were obtained via self-stimulation and collected in a sterile container following 3 to 4 days of sexual abstinence [27]. The samples were then maintained at 37 °C for 30 min, during which time they underwent liquefaction. A semen analysis was conducted in accordance with the World Health Organization’s guidelines (2010), which encompassed the assessment of motility, morphology, characteristics, and concentration. The percentage of morphologically normal spermatozoa was within the reference range in both asthenozoospermic and normozoospermic groups. However, to maintain the focus of this study on motility-related genetic factors, detailed morphology data were not included in the main text.

The sperm samples were fractionated using Percoll media gradients of 45 percent and 90 percent, followed by centrifugation at 1000× *g* for 20 min. Subsequently, the pellets were gathered and subjected to two rounds of washing with sperm-washing media to eliminate any remaining Percoll residues. Following this, the spermatozoa were retrieved by centrifugation at a force of 670× *g* for 10 min. Subsequently, before DNA extraction, the spermatozoa pellet was stored at −20 °C until the next use.

### 2.2. Sperm mtDNA Extraction

Mitochondrial DNA (mtDNA) was extracted using a commercially available kit, specifically the QIAamp DNA Mini Kit (QIAGEN GmbH, Hilden, Germany). Additionally, mtDNA enrichment was performed using another commercial kit, namely the REPLI-g Mitochondrial DNA Kit (QIAGEN GmbH, Hilden, Germany), following the instructions provided by the manufacturer. The isolated mitochondrial DNA (mtDNA) was stored at a temperature of −80 °C until it was ready to be utilized.

### 2.3. Polymerase Chain Reaction (PCR)

Primers created especially for the target sequence using PRIME 3 software were used in the polymerase chain reaction (PCR) to amplify the *MT-CO2* gene variant. The primers used from Integrated DNA Technologies (IDT) (Coralville, IA, USA) were based on the human mitochondrial DNA sequence accession number NC_012920 from NCBI. To prepare the PCR sample, a total volume of 30 µL was created by mixing 15 µL of the 2X my pols master mix (myPOLS Biotec-Germany), 0.7 µL of each primer (10 picomole) (Table 1), 3 µL of diluted amplified mtDNA, and 10.6 µL of nuclease-free water.

A MiniAmp thermal cycler from Applied Biosystems (Thermo Fisher Scientific, Singapore) was used to conduct the PCR. The amplification conditions are listed in Table 1. The amplified products were separated on a 2% agarose gel at 90 V for 40 min using 1X Tris-Borate-EDTA (TBE) buffer to confirm the PCR products. A UV documentation system was utilized to see the separated products, and a 100 bp DNA marker (BIO-HELIX-DM001-R500) was employed as a size reference.

### 2.4. Sequencing

The PCR products were sequenced locally at Enzyme company (Irbid, Jordan) according to the standard method of the Sanger sequence.

### 2.5. Data and Statistical Analysis

The *MT-CO2* gene was subjected to sequence analysis. This involved comparing it with the reference sequence of the *MT-CO2* gene before bioinformatic and statistical analysis. The bioinformatic analysis was based on the comparison to the wild-type sequence of the *MT-CO2* gene (GenBank accession number: NC_012920). The mt-DNA variants were determined by analyzing the sequences using Mutation Surveyor (v 3.9), FinchTV (v 1.4.0), and Uniprot UGENE (v 50.0) software. Single-nucleotide polymorphisms (SNPs) were detected using the Mitomap website, accessible at http://www.mitomap.org (accessed on 3 August 2025). The harmful impact of a missense mutation was evaluated using the PolyPhen-2 software, which can be accessed at the following website: http://genetics.bwh.harvard.edu/pph2/ (accessed on 3 August 2025). The genotypes and allele frequencies of the asthenozoospermic and fertile groups were analyzed using the chi-square and Fisher’s exact tests, respectively. The allele frequencies of the asthenozoospermic and normozoospermic groups were assessed to determine the odds ratios (ORs) and 95% confidence intervals (CIs). A *p*-value was deemed to be statistically significant if it was less than or equal to 0.05. The statistical analyses were conducted using the GraphPad Prism 6 software.

## 3. Results

The individuals involved in this research were categorized into two distinct cohorts: a control cohort (Normozoospermic, *n* = 77) exhibiting motility equal to/greater than 40%, and a case–cohort (Asthenozoospermic, *n* = 119) characterized by a motility range of 0 to <40%. There was no statistically significant disparity in the average age observed between the infertile and fertile groups, with a *p*-value greater than 0.05. The statistical analysis revealed a significant difference in the mean motility between the compared groups (*p* < 0.05), as presented in Table 2.

### 3.1. Gel Electrophoresis

In this study, agarose gel electrophoresis was used to detect the PCR fragments of the *MT-CO2* gene (820 bp) on 2% agarose gel (Figure 1).

### 3.2. Genotypes Analysis

The sequencing analysis of the *MT-CO2* gene was conducted for normozoospermic and asthenozoospermic men and revealed three novel genetic variants in 9 asthenozoospermia patients (Table 3), where two missense mitochondrial variants and one synonymous variant (Figure 2 and Table 3), were not recorded in the database. At the same time, these variants were absent in normozoospermic individuals. Notably, normozoospermic men harboring these variants exhibited sperm motility values close to the lower limit of the normal range (40–45%), indicating a possible subclinical effect of these variants on sperm motility. In addition, the presence of missense mutations in the normozoospermic individuals can be attributed to the low load of mutated Heteroplasmy mtDNA or suggesting that the presence of the variant alone may not be sufficient to cause the phenotype alteration, and proposing that these variants could be modifying factors, whose phenotypic expression might depend on the presence of other genetic and nongenetic factors.

In particular, the m: 7997 G>A variant was found in three asthenozoospermic patients. This variant substituted the valine in position 138 for isoleucine (Figure 2A), and another novel variant in m:8222 T>A sample substituted the leucine in position 213 for isoleucine (Figure 2B). The other alteration, m.7846 A>G, was found in five asthenozoospermic patients. This variant is classified as synonymous and coded for threonine in position 87. Interestingly, the results showed that the motility percentage of mutated samples ranged from 3–25%.

PolyPhen-2 analysis for m.7997 predicted that this variant is “probably damaging”, with a score of 0.876 on HumDiv models, and possibly damaging, with a score of 0.928 on HumVar models. In the third variant, m.8222 T<A is probably damaging with scores of 0.928 while possibly damaging with a score of 0.989 (Figure 3 and Figure 4). The impact of the three novel substitutions on the secondary and three-dimensional protein structure was performed using several websites, such as NCBI structure and PolyPhen-2.

A collective count of twenty-three single-nucleotide polymorphisms (SNPs) within the *MT-CO2* gene was identified in both the asthenozoospermic and normozoospermic groups, occupying distinct nucleotide positions. These SNPs have been previously documented in the NCBI National Center for Biotechnology Information (https://ncbi.nlm.nih.gov/ accessed on 17 September 2025) and the Human Mitochondrial Database (http://www.mitomap.org accessed on 17 September 2025). A total of four single-nucleotide polymorphisms (SNPs), specifically missense variants, were identified. The amino acid alteration in multiple mitochondrial subunits is accountable. rs386420037 G>A (Val 90 Ile), rs879119797 G>A (Val 74 Ile), rs373105186 G>A (Asp 92 Asn), rs1116904 G>A (Ala 148 Thr). On the other hand, nineteen synonymous SNPs were recorded in this study.

To ascertain the potential correlation between the variations in *MT-CO2* and infertility, we conducted a comparative analysis of the genotypes and allele frequencies between the groups of individuals with infertility (referred to as the case group) and those without infertility (referred to as the control group). The results of this analysis can be found in Table 4.

To assess the potential impact of sequence variations on the hydrophobicity and transmembrane domain of Cytochrome c oxidase subunit 2 (MTCO2) protein were employed by used the online tool PolyPhen-2 (http://genetics.bwh.harvard.edu/pph2/) and Protter https://wlab.ethz.ch/protter/start/ (accessed on 3 August 2025) as shown in Figure 3.

The 3D structure of MT-CO2 protein and missense variants in complex IV by Uniprot website (https://www.uniprot.org/uniprotkb/A0A346LZQ7/external-links) (accessed on 3 August 2025) (Figure 4).

## 4. Discussion

In this study, we explored the relationship between *MT-CO2* gene variants and asthenozoospermia in a group of Jordanian men. We identified 3 novel variants (m.8222T>A, m.7997G>A, and m.7846A>G) in addition to 23 variants that had already been reported in the literature. Of particular note, the rs1556423316 T>C variant showed a significant association with asthenozoospermia (*p* < 0.05). These findings add to the growing evidence that mitochondrial DNA (mtDNA) mutations can play an important role in the development of male infertility. The present findings further substantiate the hypothesis that COXII activity plays a pivotal role in regulating sperm motility by maintaining efficient mitochondrial ATP production. Genetic variations or functional impairments in the *MT-CO2* gene may disrupt oxidative phosphorylation, thereby diminishing energy availability required for flagellar movement and ultimately contributing to the development of asthenozoospermia. According to previous studies, alterations in COXII structure or function have been associated with impaired mitochondrial respiratory chain activity and reduced sperm motility [10,22].

Several other mitochondrial genes encoding components of the electron transport chain (ETC), such as MT-ATP6, MT-CO1, MT-CO3, MT-ND5, and MT-CYB, have been previously reported to harbor pathogenic variants associated with defective oxidative phosphorylation and impaired sperm motility. These mutations can disrupt ATP synthesis and compromise mitochondrial energy production, which is essential for sperm flagellar movement and fertilization capacity. According to previous studies, dysfunctions in these genes may act independently or in combination to exacerbate mitochondrial respiratory chain deficiencies [18,19,28,29]. In this context, the *MT-CO2* variants identified in the present study may contribute either individually or synergistically to the overall impairment of ETC function, thereby influencing sperm motility and male fertility.

Sperm motility is an energy-dependent process that relies on ATP, which is produced by OXPHOS reactions. This energy allows sperm to move through the female reproductive tract and ultimately penetrate the oocyte. During the OXPHOS process, electrons are transferred along the electron transport chain, releasing energy that drives proton pumping across the inner mitochondrial membrane. The resulting gradient powers ATP synthesis, which is critical for motility and fertilization. The *MT-CO2* gene encodes cytochrome c oxidase subunit II (COX2), an essential component of the electron transport chain. COX2 is responsible for transferring electrons from cytochrome c to molecular oxygen, the final electron acceptor. Any disruption in this step can reduce the efficiency of ATP production. Mutations in *MT-CO2* can therefore impair mitochondrial function, lower ATP levels, and, in turn, reduce sperm motility [30,31].

Two of the novel variants identified here, m.8222T>A and m.7997G>A, are missense mutations that are likely to interfere with the structure or function of COX2. Changes like these in the respiratory chain often disrupt oxidative phosphorylation and ATP synthesis, which can reduce sperm motility. The mutation in the *MT-CO2* gene leads to a complete or partial respiratory chain deficit, as observed in cells containing homoplasmic *MT-CO2* mutations [32,33]. The third variant, m.7846A>G, is synonymous and does not change the amino acid sequence. However, synonymous mutations can still affect mitochondrial function indirectly by altering mRNA stability, splicing, or translation efficiency.

Among all the variants, rs1556423316 T>C stood out as being significantly associated with asthenozoospermia (*p* < 0.05). This change may compromise COX2 function, disrupt the efficiency of the electron transport chain, and decrease ATP production, all of which would negatively affect sperm motility. Although the rs1556423316 mutation was identified in both asthenozoospermic and normozoospermic men, its frequency was markedly higher in the asthenozoospermic group (*p* < 0.05), suggesting a possible quantitative or heteroplasmic effect. Such findings indicate that the pathogenic impact of mitochondrial variants may depend on the proportion of mutant to wild-type mtDNA molecules within spermatozoa. It is plausible that in normozoospermic individuals, the mutant load remains below the functional threshold required to impair mitochondrial respiration and ATP synthesis. Conversely, in asthenozoospermic men, cumulative oxidative stress or gene–environment interactions may exacerbate mitochondrial dysfunction, leading to impaired sperm motility. This interpretation aligns with the heteroplasmic threshold model commonly described in mitochondrial genetics. The strong link between rs1556423316 T>C and asthenozoospermia in our population highlights the value of population-specific genetic screening. Incorporating mitochondrial DNA analysis into infertility diagnostics could allow earlier detection of at-risk individuals and open the door to more tailored treatment strategies. We also detected 23 previously known SNPs within *MT-CO2*, including four missense variants: rs386420037 (Val 90 Ile), rs879119797 (Val 74 Ile), rs373105186 (Asp 92 Asn), and rs1116904 (Ala 148 Thr). These variants were found in both asthenozoospermic and normozoospermic men. While they may not directly cause asthenozoospermia, they could act as modifying factors, contributing to mitochondrial dysfunction when combined with other genetic or environmental influences. In addition, several studies demonstrated a notable correlation between the presence of extensive mtDNA deletions and the prevalence of male infertility in different populations [10,16,23,34]. For instance, previous research has largely focused on other mtDNA genes, such as MT-ATP6 and MT-ND5. Several polymorphisms in *MT-CO1*, *MT-CO2*, *MT-CO3*, *MT-Cytb*, *MT-ATPase6*, and *8* showed a possible role to play in disrupting ATP generation, resulting in spermatogenesis arrest and decreased sperm motility [28,35].

A recent study reported a missense variant (m.7805 G>A) in the *MT-CO2* gene of a Tunisian asthenozoospermic individual, which resulted in the substitution of a highly conserved isoleucine residue at position 174 with valine [36]. Furthermore, another study reported four substitution alterations in the *MT-CO2* gene that have synonymous effects, namely m.7768 A>G, m.7771A>G, m.7789G>A, and m.8206 G>A [28].

Previously, Thargaraj et al. (2003) [29] reported a novel 2 bp deletion (nucleotides 8195 and 8196) in the cytochrome oxidase subunit II gene in Oligoasthenoteratozoospermia (OAT) in Indian men. The appearance of the truncated protein and interpretation of this as a decrease in sperm motility were due to a 2-nucleotide deletion in the *MT-CO2* gene [29]. Another study on the Ivory Coast population identified an unreported mitochondrial mutation (m.8027G>A) in the *MT-CO2* gene of asthenozoospermia, resulting in the conversion of alanine to threonine at position 148. This substitution changes the amino acid alanine to threonine in a transmembrane region of mitochondria and introduces a polar group, which can affect the local hydrophobicity/hydrophilicity of the protein; this increased polarity may disrupt the favorable hydrophobic interactions between the original alanine residue and the lipid bilayer. Also, the hydroxyl group in threonine can participate in hydrogen bonding interactions, potentially influencing the protein’s tertiary structure and protein’s interaction with the lipid bilayer, in addition to altering the protein’s conformation within the membrane [37].

By examining *MT-CO2*, our study underscores its importance in mitochondrial function and sperm motility. Few studies have investigated *MT-CO2* variants in specific populations, making these results particularly relevant for the Jordanian context. They are also consistent with global findings that link mitochondrial dysfunction to poor sperm motility [22,23,38,39]. Functional experiments are now needed to determine how the newly reported variants, particularly m.8222T>A and m.7997G>A, affect COX2 activity. Studies using in vitro models could clarify their impact on oxidative phosphorylation and ATP generation. Future work should also examine how these mutations interact with oxidative stress markers, which could help explain the wider effects of mitochondrial dysfunction on sperm motility.

Finally, expanding this research to larger and more diverse groups will be important to confirm these findings and evaluate their broader relevance. New technologies such as single-cell mitochondrial sequencing may provide a deeper understanding of the complex genetic and epigenetic factors underlying asthenozoospermia.

## 5. Conclusions

The current study reported three novel variants in nine asthenozoospermic patients (m.7846, m.7997, and m.8222), which were not registered in the database, and four known missense variants. These variants are proposed to be associated with asthenozoospermic infertility in the Jordanian population.

## Figures and Tables

**Figure 1 cimb-47-00901-f001:**
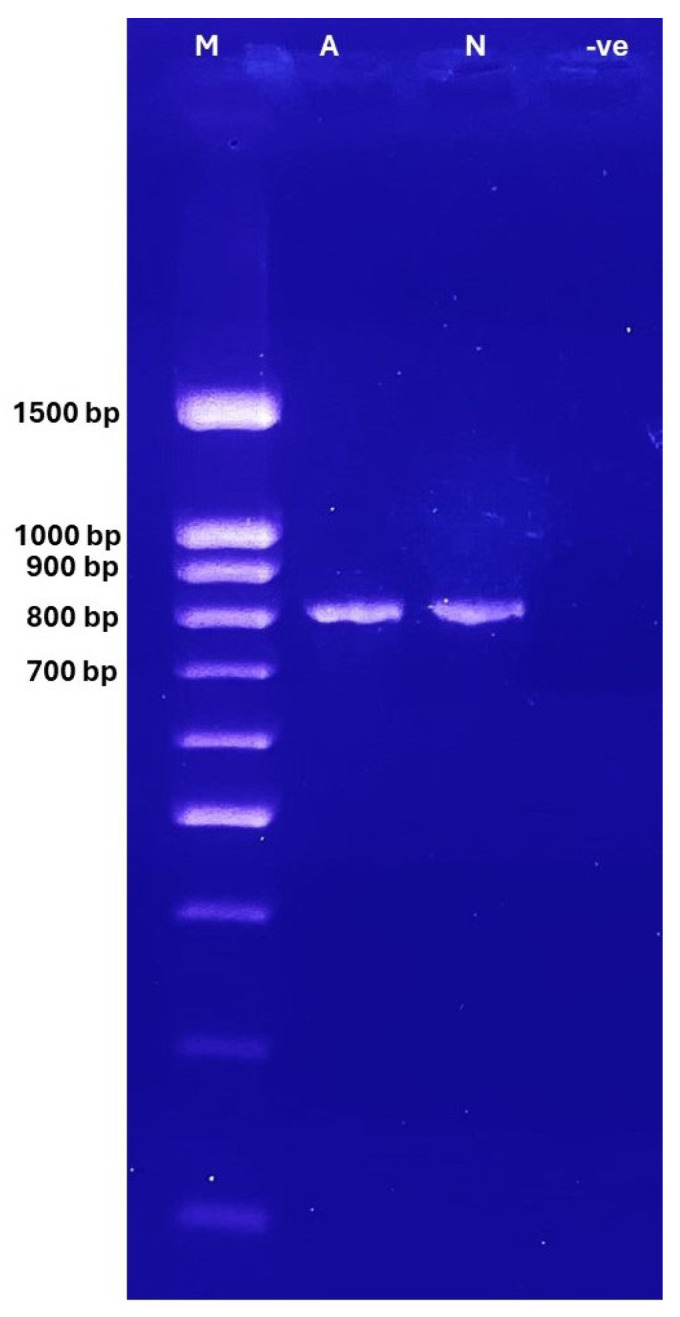
PCR products of the mt-COX2 gene (820 bp) on agarose gel electrophoresis, 2% under UV light; M: ladder 100–1500 (BIO-HELIX). Lane M = Ladder, Lane A = asthenozoospermic sample, Lane N = normozoospermic samples, and Lane-ve: negative control.

**Figure 2 cimb-47-00901-f002:**
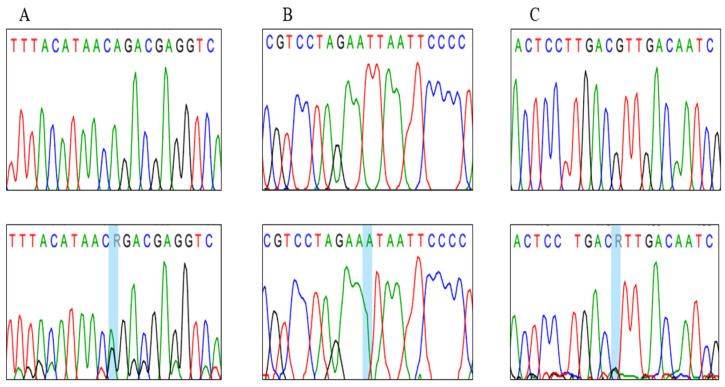
Sequence chromatograms of novel variants recorded in the *MT-CO2* gene, (**A**) m.7997 heteroplasmy variant (allele G>A), (**B**) m.8222 homoplasmy variants (allele T> A), and (**C**) m.7997 G>A heteroplasmy variant (allele G>A).

**Figure 3 cimb-47-00901-f003:**
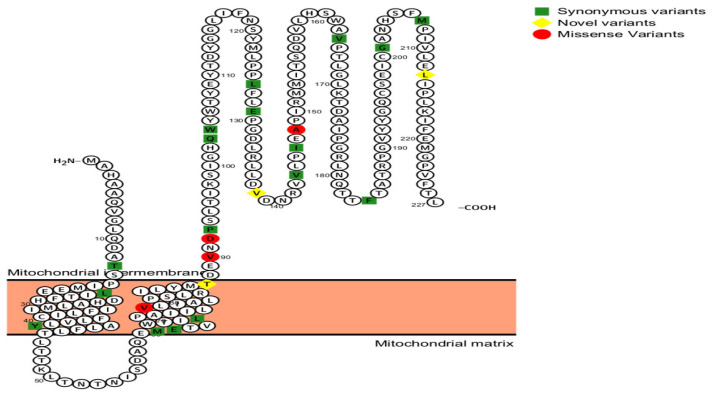
Predicted the secondary structure of human MT-CO2 protein sequence and the position of missense and synonymous variants in the transmembrane structures created in the protter website; red color: missense variants, yellow color: novel, and green color: synonymous SNPs (http://wlab.ethz.ch/protter/start/) (accessed on 3 August 2025).

**Figure 4 cimb-47-00901-f004:**
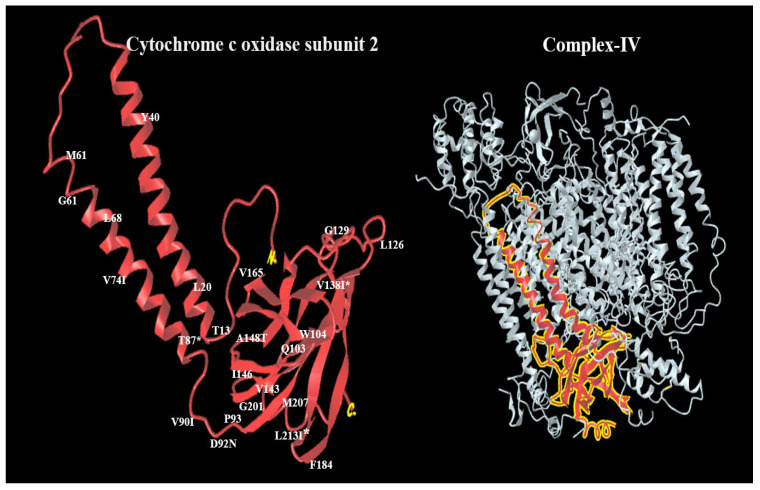
Three-dimensional structure of Cytochrome c oxidase subunit 2, which is part of Complex IV in the mitochondrial electron transport chain, labeled with recorded variants at different positions, * novel variant.

**Table 1 cimb-47-00901-t001:** Oligonucleotide primers and cycling conditions used for PCR amplification of the *MT-CO2* gene.

Primer	Sequences (5′ → 3′)	Cycling Conditions		Product Size (bp)
Forward	5-CATGGCCTCCATGACTTTTT-3	Initial denaturation 94 °C 5 min Denaturation 94 °C 40 s Annealing 62 °C 40 s Extension 72 °C 45 s Final extension 72 °C 5 min	} 35 cycles	820
Reverse	5-GTTAGCTTTACAGTGGGCTC-3			

**Table 2 cimb-47-00901-t002:** Comparison of the semen analysis parameters (age and motility) between the asthenozoospermic and normozoospermic samples.

Parameter	Age (Mean)	Motility (Mean)	*p*-Value *t*-Test
Asthenozoospermic (*N* = 119)	34.3	12.58	0.001
Normozoospermic (*N* = 77)	32.66	62.10	

**Table 3 cimb-47-00901-t003:** Genotype and amino acid change for novel variant in *MT-CO2* polymorphisms between asthenozoospermic and normozoospermic males.

Position	Codon Change	Amino Acid Change	Homoplasmy/Heteroplasmy/	No. of Individuals	Polyphen2 Prediction
m.7846 A>G	ACA>ACG	T87T	Heteroplasmy (A, G)	5 Asthenozoospermic 2 Normozoospermic	-
m.8222 T>A	TTA>TAA	L213I	Homoplasmy (A, A)	1 Asthenozoospermic 1 Normozoospermic	Possibly damaging
m.7997 G>A	GTT>ATT	V138I	Homoplasmy (G, G) Heteroplasmy (G, A)	3 Asthenozoospermic 1 Normozoospermic	Probably damaging

**Table 4 cimb-47-00901-t004:** Genotype frequency of *MT-CO2* polymorphisms between asthenozoospermic and normozoospermic males.

rs ID	Contig Position	Codon Change	Amino Acid Change	Type of Variant	Homoplasmy/Heteroplasmy/	Frequency of Mutation Asthenozoospermic (N) (%)	Frequency of Mutation Normozoospermic (N) (%)	*p* Value **(Chi-Square Test).**
rs28358879	7624T>A	ACT > ACA	Thr13	Synonymous	Homoplasmy (A, A)	1/118 (0.85%)	0/77 (0%)	0.4200
rs1556423316	7645T>C	CTT > CTC	Leu20	Synonymous	Homoplasmy (C, C) Heteroplasmy (T, C)	7/109 (6.42%) 3/109 (2.75%)	14/58 (24.14%) 5/58 (8.62%)	0.0072
rs1556423330	7705T>C	TAT > TAC	Tyr40	Synonymous	Homoplasmy (C, C)	3/116 (2.59%)	0/77 (0%)	0.1603
rs41534044	7768A>G	ATA > ATG	Met61	Synonymous	Homoplasmy (G, G)	3/116 (2.59%)	0/77 (0%)	0.1603
rs368038563	7771A>G	GAA > GAG	Glu62	Synonymous	Homoplasmy (G, G)	2/117 (1.71%)	0/77 (0%)	0.2529
rs386829014	7789G>A	CTG > CTA	Leu68	Synonymous	Homoplasmy (A, A)	2/117 (1.71%)	1/76 (1.31%)	0.8315
rs879119797	7805G>A	GTC > ATC	Val74Ile	Missense	Homoplasmy (A, A)	3/116 (2.59%)	2/75 (1.33%)	0.9736
rs386420037	7853G>A	GTC > ATC	Val90Ile	Missense	Homoplasmy (A, A)	2/117 (1.71%)	1/76 (1.31%)	0.8315
rs373105186	7859G>A	GAT > AAT	Asp92Asn	Missense	Homoplasmy (A, A)	1/118 (0.85%)	1/76 (1.31%)	0.7552
rs879034269	7864C>T	CCC > CCT	Pro93	Synonymous	Homoplasmy (C, C)	1/118 (0.85%)	0/77 (0%)	0.4200
rs386829020	7894A>G	CAA> CAG	Gln103	Synonymous	Heteroplasmy (A, G)	1/118 (0.85%)	0/77 (0%)	0.4200
rs1603221196	7897G>A	TGG> TGA	Trp104	Synonymous	Homoplasmy (A, A)	1/118 (0.85%)	0/77 (0%)	0.4200
rs373420717	7961T>C	TTA > CTA	Leu126	Synonymous	Homoplasmy (C, C)	1/118 (0.85%)	0/77 (0%)	0.4200
rs879023568	7963A>G	TTA > TTG	Leu126	Synonymous	Homoplasmy (G, G)	2/117 (1.71%)	0/77 (0%)	0.2529
rs878954138	7972A>G	GAA> GAG	Glu129	Synonymous	Heteroplasmy (A, G)	0/119 (0%)	1/76 (1.31%)	0.2126
rs879223416	8014A>G	GTA > GTT	Val143	Synonymous	Homoplasmy (G, G)	1/118 (0.85%)	4/73 (5.48%)	0.0590
rs879039143	8023T>C	ATT > ATC	Ile146	Synonymous	Homoplasmy (C, C)	1/118 (0.85%)	0/77 (0%)	0.4200
rs1116904	8027G>A	GCC> ACC	Ala148Thr	Missense	Homoplasmy (A, A) Heteroplasmy (G, A)	0/118 (0%) 1/118 (0.85%)	1/76 (1.31%) 0/76 (0%)	0.7552
rs371628304	8080C>T	GTC > GTA	Val165	Synonymous	Homoplasmy (T, T)	1/118 (0.85%)	1/76 (1.31%)	0.7552
rs879043235	8137C>T	TTC > TTT	Phe184	Synonymous	Homoplasmy (T, T)	1/118 (0.85%)	0/77 (0%)	0.4200
rs28651339	8188A>G	GGA > GGG	Gly201	Synonymous	Homoplasmy (G, G) Heteroplasmy (A, G)	1/118 (0.85%) 0/118 (0%)	0/76 (0%) 1/76 (1.31%)	0.3340
rs28358883	8206G>A	ATG > ATA	Met207	Synonymous	Homoplasmy (A, A) Heteroplasmy (G, A)	4/114 (3.51%) 1/114 (0.88%)	1/76 (1.31%) 0/76 (0%)	0.4852
rs386829020	7894 A>G	CAA> CAG	Gln103	Synonymous	Heteroplasmy (A, G)	1/118 (0.85%)	0/77 (0%)	0.4253

## Data Availability

The original contributions presented in this study are included in the article. Further inquiries can be directed to the corresponding author.

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
