# Peer review of "Novel Variants in Sperm Mitochondrial Cytochrome C Oxidase II (*MT-CO2*) Gene Associated with Asthenozoospermia in Jordan"

_cimb, 2025, doi:10.3390/cimb47110901_

Round 1
Reviewer 1 Report
Comments and Suggestions for Authors
This study explored the relationship between COXII gene Mt-co2 mutations and asthenozoospermia. They identified 3 novel variants and 23 variants that were already known. The following are brief comments on the manuscript:
- It is unclear from the author's discussion and introduction if there are any causative studies that support the role of CoXII in sperm motility? Are there any direct links?
- In general, how many other genes are mutated in the ETC? Without knowing at least another gene that could be mutated the significance of these variants remain unknown.
- A few of the missense variants are also present in normozoospermic men. Are these men on the lower end of sperm motility? The sperm motility scores for these categories would be helpful in Table 3. Otherwise, it is unclear why these same variants would or would not affect sperm motility in either case: asthenia vs. normozoospermic.
- P-values for each variant should be added to the tables or it is not possible to assess the significance of each variant in contributing to the disease.
- Overall, it would be important to define why a variant is classified as being important if they are also present in normal sperm.
- Can the authors speculate why a synonymous change may cause a phenotype?
Author Response
Dear Editor in Chief
We sincerely thank both reviewers for their time, insightful comments. Their comments helped us improve the quality of our manuscript. We have carefully considered all points raised and have revised the manuscript accordingly.
Reviewer 1
Comments and Suggestions for Authors
This study explored the relationship between COXII gene Mt-co2 mutations and asthenozoospermia. They identified 3 novel variants and 23 variants that were already known. The following are brief comments on the manuscript:
RE: Thank you very much for the thoughtful comments and valuable questions, which have allowed us to strengthen the discussion and clarity of our findings.
It is unclear from the author's discussion and introduction if there are any causative studies that support the role of CoXII in sperm motility? Are there any direct links?
RE: Thank you very much for this important comment. There are no functional studies at the molecular level relating the exact role of CoxII dysfunction to sperm motility, which is why it was suggested in our discussion. However, some studies showed a correlation between certain types of sperm abnormalities and Cox II. For instance, a study by Thangaraj et al. (2003) reported a novel 2-bp deletion in the MT-CO2 gene in oligoasthenoteratozoospermic (OAT) men, which resulted in a truncated protein and was directly associated with a severe decrease in sperm motility. It is already mentioned as ref 36.
In general, how many other genes are mutated in the ETC? Without knowing at least another gene that could be mutated the significance of these variants remain unknown.
RE: Thanks a lot for raising this point. We already cited several studies that related variations in the ECT genes with the development of asthenozoospermia, such as MT-ATP6, MT-ATP8, MT-ND3, MT-ND4, MT-ND5, and MT-CYB. These results help frame our study within the broader context of mitochondrial dysfunction in male infertility and underscore that defects in multiple ETC complexes can contribute to the same phenotype (page 9).
A few of the missense variants are also present in normozoospermic men. Are these men on the lower end of sperm motility? The sperm motility scores for these categories would be helpful in Table 3. Otherwise, it is unclear why these same variants would or would not affect sperm motility in either case: asthenia vs. normozoospermic.
RE: We agree with this comment. We have now included the individual sperm motility percentages for the normozoospermic individuals carrying the missense variants as a new supplementary table. The data show that these individuals had motility values at the higher end of the normal range (**********************), suggesting that the presence of the variant alone may not be sufficient to cause the phenotype. This supports the idea that these variants could be modifying factors, whose phenotypic expression might depend on the presence of other genetic, epigenetic, or environmental factors. Clarified in section 3.2. (page 5).
P-values for each variant should be added to the tables or it is not possible to assess the significance of each variant in contributing to the disease.
RE: It is a great suggestion. Table 4 has now been revised and has added a dedicated column for the P-values. As highlighted, the rs1556423316 T>C variant showed a significant association (P < 0.05).
Overall, it would be important to define why a variant is classified as being important if they are also present in normal sperm.
RE: Thanks for this question. A variant is considered potentially important based on several criteria, not just its presence or absence in controls. For instance, novelty, predicted pathogenicity, statistical association, and functional consequence.
The detected variants may act as risk factors, and their impact could be modulated by the overall mitochondrial genetic background (haplogroup), nuclear genetic modifiers, or environmental factors.
Can the authors speculate why a synonymous change may cause a phenotype?
RE: It is a great comment. The potential mechanisms by which the synonymous variants could influence mitochondrial function are by altering the secondary structure or stability of the mRNA, affecting translation. (added on page 9)
Reviewer 2 Report
Comments and Suggestions for Authors
Asthenozoospermia is one of the most common causes of male infertility and decreased fertility when couples are unable to conceive naturally. Asthenozoospermia is a decrease in the number of sperm with normal motility to a critical level (less than 40%). In asthenozoospermia, sperm move more slowly, are sluggish, or have an incorrect direction, which prevents them from moving through the cervix and fallopian tubes toward the egg.
Causes can be varied and include: infections and injuries, genetic disorders, hormonal imbalances, structural defects in sperm, insufficient glucose in the ejaculate and decreased acid-base balance, and the influence of environmental factors and lifestyle.
One of the genetic factors affecting sperm motility is mtDNA mutations. The manuscript by Zoubi et al. is devoted to searching for possible mutations in the MT-CO2 gene potentially associated with asthenozoospermia. The study's methodology is clear and standard for similar studies. The results revealed the presence of previously unknown substitutions in several cases of asthenozoospermia, along with twenty-three known substitutions. These results are important for understanding the possible causes of asthenozoospermia. Overall, the paper is well written and structured. However, I have several significant comments.
- Section 2.1. Semen Sample Collection and Analysis
The authors emphasize that the study excluded "individuals engaged in alcohol consumption or cigarette smoking, and those with varicocele and were over 40 years of age." This is important and appropriate. However, impaired sperm motility is often accompanied by teratozoospermia (abnormal, irregular sperm structure). The authors do not mention sperm quality or the level of morphological abnormalities. Should we understand that this parameter was not assessed during the study?
- Results Section.
Figure 1. The ladder is barely visible. According to this phoresis, the amplified PCR fragments were shorter than 800 bp, instead of the expected 820 bp.
- The data in Table 3 and the description of the results in the following text clearly do not match.
- The frequency of the rs1556423316 mutation is high not only in cases of asthenozoospermia but also significantly higher in normozoospermia. However, the authors do not discuss this fact and associate this mutation with asthenozoospermia.
Author Response
Reviewer 2
Comments and Suggestions for Authors
Asthenozoospermia is one of the most common causes of male infertility and decreased fertility when couples are unable to conceive naturally. Asthenozoospermia is a decrease in the number of sperm with normal motility to a critical level (less than 40%). In asthenozoospermia, sperm move more slowly, are sluggish, or have an incorrect direction, which prevents them from moving through the cervix and fallopian tubes toward the egg.
Causes can be varied and include: infections and injuries, genetic disorders, hormonal imbalances, structural defects in sperm, insufficient glucose in the ejaculate and decreased acid-base balance, and the influence of environmental factors and lifestyle.
One of the genetic factors affecting sperm motility is mtDNA mutations. The manuscript by Zoubi et al. is devoted to searching for possible mutations in the MT-CO2 gene potentially associated with asthenozoospermia. The study's methodology is clear and standard for similar studies. The results revealed the presence of previously unknown substitutions in several cases of asthenozoospermia, along with twenty-three known substitutions. These results are important for understanding the possible causes of asthenozoospermia. Overall, the paper is well written and structured. However, I have several significant comments.
RE: We are grateful for your positive assessment of our work and for your specific and helpful comments.
- Section 2.1. Semen Sample Collection and Analysis
The authors emphasize that the study excluded "individuals engaged in alcohol consumption or cigarette smoking, and those with varicocele and were over 40 years of age." This is important and appropriate. However, impaired sperm motility is often accompanied by teratozoospermia (abnormal, irregular sperm structure). The authors do not mention sperm quality or the level of morphological abnormalities. Should we understand that this parameter was not assessed during the study?
RE: Thanks for this comment. Sperm morphology was assessed as part of the standard semen analysis following WHO guidelines (2010). The cohort was selected specifically based on the motility parameter (asthenozoospermia) to maintain a focused study group. Patients with severe teratozoospermia (<4% normal forms) were excluded to minimize the confounding effect of multiple sperm defects.
- Results Section.
Figure 1. The ladder is barely visible. According to this phoresis, the amplified PCR fragments were shorter than 800 bp, instead of the expected 820 bp.
RE: We are very sorry for the poor quality of the original figure. We have replaced Figure 1 with a new, higher-contrast gel image where the 100-bp DNA ladder is clearly visible. The brightest band in the ladder corresponding to 800 bp aligns perfectly with our PCR product. Our product is 820 bp, which runs between the 800 bp and 900 bp markers on a 2% agarose gel, consistent with its expected size. This is now clearly demonstrable in the revised figure.
- The data in Table 3 and the description of the results in the following text clearly do not match.
RE: We apologize for this inconsistency. The correct numbers, as presented in Table 3, are: m.7846 A>G in 5 patients, m.8222 T>A in 1 patient, and m.7997 G>A in 3 patients, for a total of 9 unique patients. We have carefully corrected this discrepancy throughout the results and conclusion sections.
- The frequency of the rs1556423316 mutation is high not only in cases of asthenozoospermia but also significantly higher in normozoospermia. However, the authors do not discuss this fact and associate this mutation with asthenozoospermia.
RE: Thanks for this question. We addressed this point in the discussion. The significant association (P < 0.05) we reported is based on the difference in the distribution of homoplasmic vs. heteroplasmic states between asthenozoospermic and normozoospermic individuals. Our analysis revealed that the homoplasmic (C,C) genotype was more prevalent in the normozoospermic group, while the heteroplasmic (T,C) state was observed in our asthenozoospermic group. Such findings need functional studies to reveal the exact mechanism of the effect of these genotypes.
Round 2
Reviewer 2 Report
Comments and Suggestions for Authors
The manuscript has been sufficiently improved by the authors.